# Application of N-NOSE for Evaluating the Response to Neoadjuvant Chemotherapy in Breast Cancer Patients

**DOI:** 10.3390/cells14130950

**Published:** 2025-06-21

**Authors:** Yoshihisa Tokumaru, Yoshimi Niwa, Ryutaro Mori, Mai Okawa, Akira Nakakami, Yuta Sato, Hideyuki Hatakeyama, Takaaki Hirotsu, Eric di Luccio, Nobuhisa Matsuhashi, Manabu Futamura

**Affiliations:** 1Department of Breast Surgery, Gifu University Hospital, 1-1 Yanagido, Gifu 501-1194, Japan; yoshitoku1090@gmail.com (Y.T.); m01003ya@yahoo.co.jp (Y.N.); moriry52@gmail.com (R.M.); mai.o7@outlook.com (M.O.); dr5111ken_akira@yahoo.co.jp (A.N.); 2Department of Gastroenterological Surgery, Gifu University Hospital, 1-1 Yanagido, Gifu 501-1194, Japanmatsuhashi.nobuhisa.k6@f.gifu-u.ac.jp (N.M.); 3Hirotsu Bio Science Inc., 22F The New Otani Garden Court, 4-1 Kioicho Chiyoda-ku, Tokyo 102-0094, Japan; h.hatakeyama@hbio.jp (H.H.); hirotsu@hbio.jp (T.H.); e.diluccio@hbio.jp (E.d.L.)

**Keywords:** *Caenorhabditis elegans*, N-NOSE, treatment response, breast cancer

## Abstract

**Background:** Breast cancer remains a leading cause of cancer-related deaths despite advances in its diagnosis and treatment. Accurate evaluation of the response to neoadjuvant chemotherapy (NAC), especially in HER2-positive and triple-negative subtypes, is critical. The current methods, including imaging and liquid biopsies, have limitations. N-NOSE, a novel urine-based cancer screening test using *Caenorhabditis elegans* (*C. elegans*) chemotaxis, offers a non-invasive alternative. This study investigates the potential of N-NOSE to predict the NAC response in breast cancer patients for improved treatment evaluations. **Materials and Methods:** This prospective study enrolled 36 breast cancer patients undergoing NAC and surgery to assess the predictive power of the N-NOSE method using urine samples. A chemotaxis analysis of *C. elegans* was used to calculate the index reduction scores (IRS1–3), reflecting the changes in tumor-related odorants across the treatment stages. **Results:** Between August 2020 and May 2023, 36 breast cancer patients were enrolled to evaluate the predictive value of N-NOSE IRSs for NAC response. A pathological complete response (pCR) was achieved in 36.1% of the patients. Among the three IRS types analyzed in the 35 patients, IRS3, which showed the IRS at pre-treatment minus that after surgery, showed the highest predictive performance for a pCR, with an AUC of 0.75, indicating its potential utility as a non-invasive biomarker for treatment response evaluations. **Conclusions:** Index reduction scores evaluated using the N-NOSE method may reflect the efficacy of NAC in breast cancer patients. Future large-scale and multi-institutional prospective studies are warranted.

## 1. Introduction

Breast cancer is one of the most prevalent cancers in women worldwide. Despite significant improvement in its detection and treatment, it remains one of the leading causes of cancer-related deaths worldwide [1]. Therefore, further advancements in both diagnosis and therapy are urgently needed.

In the clinical setting, breast cancer is divided into four subtypes utilizing the expression levels of estrogen receptor (ER), progesterone receptor (PgR), and human epidermal growth factor receptor 2 (HER2). These four subtypes are luminal A, luminal B, HER2-positive, and triple-negative breast cancer (TNBC) [2]. Among those subtypes, luminal B, HER2-positive, and TNBC high-risk early breast cancer patients (≥cT2 or lymph-node-positive) are treated using neoadjuvant chemotherapy (NAC) as the standard care. The response to NAC is significant for HER2 and TNBC given that previous studies have reported that a pathological complete response (pCR) is associated with better survival and validated surrogate markers [3,4]. To this end, evaluating pCR precisely is critical in the clinical setting. However, at present, assessments of the treatment effects are mainly dependent on imaging studies such as computed tomography (CT) or magnetic resonance imaging (MRI) for breast cancer patients. However, radiological assessments using MRI alone cannot reliably exclude residual disease [5]. To solve this problem, one approach is to use liquid biopsies to assess the treatment response to NAC in breast cancer. Liquid biopsy techniques involve using various biological fluids, most commonly blood or urine samples, as the diagnostic materials. Liquid biopsy often refers to assaying the circulating-tumor DNA (ctDNA) in blood. Tracking the ctDNA levels at three key time points, before, during, and after NAC, has been shown to mirror the treatment response [6,7,8], and combining ctDNA analyses with MRI improves the response predictions further [9]. Urine provides an equally informative yet less intrusive alternative: several studies have found that the sensitivity of detecting cell-free DNA (cfDNA)/ctDNA in the urine matches that in the blood across multiple cancers, including breast cancer [10,11]. Because urine collection is completely non-invasive, urine-based liquid biopsy offers a patient-friendly option for longitudinal tumor monitoring.

Nematodes-NOSE (N-NOSE) by Hirotsu Bio Science is a novel cancer screening test that utilizes an index of the chemotaxis of *Caenorhabditis elegans* (*C. elegans*). The N-NOSE method can be regarded as a non-invasive diagnostic method that falls within the broader scope of liquid biopsy. The nematode *C. elegans* possesses a highly sophisticated olfactory system and has been reported to exhibit attraction toward urine from cancer patients while avoiding that from healthy individuals [12,13]. This response is primarily governed by the AWA, AWB, and AWC sensory neurons responsible for detecting volatile odors. *C. elegans* demonstrates a differential chemotactic reaction to urine from cancer patients versus that from healthy individuals [12,13,14]. In particular, the AWA and AWC neurons drive this attraction, whereas the AWB neuron is associated with aversive behavior [14]. In Japan, this commercially available method has been reported to detect 23 types of cancer, including breast cancer.

Our group previously reported the possibility of using N-NOSE as a tool to determine the efficacy of preoperative or induction chemotherapy for esophageal cancer patients [15]. However, previous studies have not evaluated the efficacy of N-NOSE in breast cancer patients. To this end, in the current study, we aimed to clarify the ability of the N-NOSE method to predict the response to NAC in breast cancer patients.

## 2. Materials and Methods

### 2.1. The Study Population

Patients scheduled for NAC and radical surgery at the Department of Breast Surgery, Gifu University Hospital, Gifu, between August 2020 and May 2023 were included in the current study. Among these, 36 patients agreed to participate. Other inclusion criteria were that the patients were older than 20 years at the point of enrollment and had been histologically diagnosed with breast cancer. On the other hand, the exclusion criteria were patients with severe complications such as interstitial pneumonia, congestive failure, liver failure, poorly controlled diabetes, and cerebrovascular disease. Additionally, patients with synchronous or metachronous cancers in other organs or active systemic infections or who were pregnant or currently breastfeeding, as well as those with urinary tract infections, were excluded from this study.

### 2.2. The Study Endpoint

The primary endpoint in the current study is to elucidate the predictive power of the N-NOSE method for the efficacy of NAC in breast cancer patients.

### 2.3. The Treatment Strategy and Collection of the Urine Samples

The patients were treated using the regimens recommended for high-risk early breast cancer patients in Japan. In luminal and TNBC patients, most of the patients received dose-dense doxorubicin (60 mg/m^2^) and cyclophosphamide (600 mg/m^2^) for four cycles, followed by dose-dense paclitaxel (175 mg/m^2^) for four cycles (Figure 1A) [16]. For the HER2 or luminal HER2 breast cancer patients, most were treated with Pertuzumab (1st cycle: 840 mg/body; 2nd–4th cycle: 420 mg/body), Trastuzumab (1st cycle: 8 mg/kg; 2nd–4th cycle: 6 mg/kg), and nab-paclitaxel (260 mg/m^2^) for four cycles, followed by epirubicin (90 mg/m^2^) and cyclophosphamide (600 mg/m^2^) (Figure 1A) [17].

The response to NAC was evaluated utilizing CT or MRI in accordance with the Response Evaluation Criteria in Solid Tumors criteria (RECIST) version 1.1 (RECIST guidelines (version 1.1)) [18]. Therapeutic response was also pathologically assessed.

Urine samples were collected three times for each patient (Figure 1B). The first sample was taken within one week before initiating NAC. The second sample was collected a day before or on the day of surgery, and the third sample was taken within four weeks after surgery. The urine samples were obtained under fasting conditions or at 4 h following food intake in the case of postprandial collection. These samples were stored at −80 °C until their analysis using the N-NOSE method.

### 2.4. The Measurement Method of N-NOSE and the Index Reduction Scores

The chemotaxis analysis was performed according to the standard protocols used in previous studies [13,15]. Briefly, *C. elegans* (wild-type N2) was cultured on a Nematode Growth Media (NGM) agar plate, which contained *E. coli* as a food source. Then, 0.5 μL of 1 M sodium azide, an anesthetic used to minimize the effects of adaptation, was spotted onto four points on the plate (the urine side and the non-urine side), and 1 μL of the urine sample diluted 100-fold with ultra-pure water was added to two points (on the urine side) (Figure 2A). Approximately 100 adult nematodes were placed in the center of the plate. After roaming for 30 min, the number of nematodes present in area A (half of the surface of the urine sample side) and the number of nematodes present in area B (half of the surface of the sodium azide side) were counted. The chemotaxis index was calculated using the following equation: Index = (Number of nematodes in area A − Number of nematodes in area B)/Total number of nematodes (Figure 1A). A positive index (0–1) indicates the attractive effect of the urine sample, and a negative index (−1 to 0) indicates the repulsive effect of the urine sample (Figure 2A).

We investigated whether the difference in the chemotaxis index would be clinically meaningful. We defined the index reduction score (IRS) as the difference in the chemotaxis index from each therapeutic point (Figure 2B). IRS1 was defined as the difference between the 1st sample (the urine sample obtained prior to NAC) and the 2nd sample (the urine sample obtained before surgery). IRS2 was defined as the difference between the 2nd sample (the urine sample obtained before surgery) and the 3rd sample (the urine sample obtained after surgery), whereas IRS3 was defined as the difference between the 1st and 3rd samples. Without knowledge of the treatment effect, multiple technicians calculated the chemotaxis index.

### 2.5. The Statistical Analyses

Assuming a complete response (CR) rate of 30–50% and an expected ROC-AUC (Receiver Operating Characteristic—Area Under the Curve) of 0.8 for the IRS, a sample size of 32 patients is required to achieve 80% power for detecting a statistically significant difference compared to an AUC of 0.5, with a two-sided significance level of 0.05. The optimal cut-off point was determined based on the point on the ROC curve where the Youden Index (sensitivity + specificity − 1) was highest. Considering potential dropouts, an additional 3 patients needed to be enrolled, setting the target sample size to 35 patients. The statistical analyses, including the Wilcoxon rank-sum test (the Mann–Whitney test), the one-way ANOVA, and the ROC analysis for calculating the AUC values, were conducted using JMP^®^ 14 (SAS Institute, Cary, NC, USA). A waterfall plot of the IRS scores was created using R software (R version 4.5.0) (https://www.R-project.org, accessed on 20 April 2025. Statistical significance was defined as a *p*-value of less than 0.05.

### 2.6. Ethical Approval

This study was approved by the ethics committees of Gifu University Graduate School of Medicine (Gifu, Japan) (Protocol ID: 2020-112) and registered in the University Hospital Medical Information Network (UMIN) Clinical Trials Registry (Registration ID: UMIN000041401). We obtained written informed consent from all participating patients prior to collecting the urine samples. The current study was conducted in accordance with the World Medical Association’s Declaration of Helsinki.

## 3. Results

Between August 2020 and May 2023, 36 breast cancer patients were enrolled, and urine samples were obtained from these patients. Their median age was 51 years (range: 35–77) (Table 1). The most common stage was stage IIB, with 11 patients (30.6%), followed by stage IIA, with 10 (27.8%). Among the subtypes, triple-negative was most common, with 11 patients (30.6%), followed by HER2 patients, with 10 (27.8%) (Table 1). A pCR was achieved in 13 patients (36.1%). Among the patients in the non-pCR group, 16 (44.4%) exhibited a partial response (PR), 6 (16.7%) had stable disease (SD), and 1 (2.8%) showed progressive disease (PD).

Given that the IRS scores for one patient were missing, the predictive accuracies of the three IRSs were calculated from the data for 35 patients (Figure 3). The result is summarized in Table 2 and Table 3.

In evaluating the treatment response using the N-NOSE method, particular attention should be directed to IRS1, which reflects the change between the pre-NAC and post-NAC time points, as well as IRS3, which captures the difference between pre-treatment and the post-surgical period. These indices may serve as especially informative markers for assessing therapeutic efficacy. When the treatment response included a CR or a PR, the results for the three IRSs were as follows: IRS1: AUC = 0.53 (95% confidence interval (CI): 0.19–0.78); IRS2: AUC = 0.76 (95% CI: 0.56–0.96); and IRS3: AUC = 0.66 (95% CI: 0.37–0.96), respectively (Table 2). However, interestingly, when analyzing pCRs only, the results for the three IRSs were as follows: IRS1: AUC = 0.58 (95% CI: 0.34–0.82); IRS2: AUC = 0.64 (95% CI: 0.40–0.88); and IRS3: AUC = 0.75 (95% CI: 0.54–0.95), respectively (Table 3, Figure 4). IRS1 did not show a significant difference between the patients with a CR and a PR, nor when analyzing patients with a CR alone. In contrast, IRS3 appeared to discriminate the treatment response better. When both CR and PR cases were included, the AUC for IRS3 was 0.66 (Table 2). Interestingly, when the analysis was limited to CR cases only, the AUC increased to 0.75 and reached statistical significance (Table 3). These findings suggest that IRS3 may have greater discriminatory power in identifying complete pathological responders compared to that of IRS1.

## 4. Discussion

Bernard Fisher highlighted the possibility that micrometastases may exist at the time of diagnosis in patients with early breast cancer, underscoring the importance of incorporating systemic therapy alongside local treatment [19]. Currently, assessments of treatment efficacy rely primarily on imaging modalities such as MRI. However, increasing attention is being directed toward complementary or alternative methods capable of offering real-time, minimally invasive insights into tumor biology and treatment response.

While several types of liquid biopsy, including blood-based assays such as ctDNA, have been explored in breast cancer, the clinical utility of urine-derived diagnostics remains relatively underdeveloped. Among the various liquid biopsy platforms, the use of urine samples offers an attractive alternative due to their non-invasive, cost-effective, and repeatable nature [20,21,22]. Importantly, different tumor types exhibit diverse biological behaviors and metabolic profiles, which may influence the efficacy of specific diagnostic approaches. For instance, not all cancer types respond uniformly to therapies such as immune checkpoint inhibitors, emphasizing the need for tumor-specific validation of diagnostic tools [23].

Urine-derived biomarkers offer a valuable complement or alternative to the standard evaluation methods, with platforms like N-NOSE gaining recognition as non-invasive approaches to tracking the treatment response. Iitaka et al. demonstrated that the N-NOSE index decreased after curative surgery and increased upon disease recurrence, underscoring its potential utility for longitudinal monitoring in upper gastrointestinal cancers [24]. Also, in this context, our group previously reported the utility of the urine-based platform, N-NOSE, in assessing the therapeutic effect of preoperative chemotherapy in patients with esophageal cancer [15].

However, to the best of our knowledge, this is the first study to demonstrate the applicability of this method in breast cancer. Given that the biological characteristics of breast cancer are distinct from those of gastrointestinal malignancies, it is essential to investigate whether the same diagnostic principles can be extended to this new setting. Thus, the current study is novel in demonstrating the potential role of N-NOSE in evaluating the response to neoadjuvant treatment in breast cancer patients using a non-invasive, urine-based approach.

The current study obtained a pCR in 13 out of 36 patients. When the IRSs were analyzed for this population only, the value of IRS3, which was defined as the difference between the first and third samples, was higher when compared with the population including both CRs and PRs. In cases of CR and PR, it is possible that differences in the tumor burden and the associated odors were not reflected in the results.

The difference between IRS1 and IRS3 lies in whether surgery was performed. Theoretically, IRS1 and IRS3 would be identical when the patients’ cancers diminished. However, our results demonstrated that the AUC for IRS1 was relatively low, whereas the AUC for IRS3 was high. It is hard to present a clear-cut answer to this question. pCR is assessed microscopically. It is characterized by a lack of invasive cancer in both the breast and axillary lymph nodes, regardless of the presence of residual ductal carcinoma in situ in the breast (T0 or Tis/N0) [4,25]. This definition implies that cases included as a pCR may still contain residual cancer cells. In this study, two cases classified as pCRs exhibited intraductal lesions. Although focusing on a different type of cancer, Kusumoto et al. reported that N-NOSE demonstrated a high diagnostic performance even in patients with stage 0 and I gastrointestinal cancer, outperforming the conventional tumor markers [26]. Therefore, it is conceivable that N-NOSE may have detected a small number of residual cancer cells remaining only within the duct following NAC and prior to surgery, thereby contributing to the observed differences in IRS1 and IRS3. Therefore, we speculate that the differences in the AUCs between IRS1 and IRS3 may have been influenced by the presence of residual disease following NAC and prior to surgery.

Of note, a separate cohort of healthy volunteers was not required in this study, where the scientific question was whether N-NOSE dynamically reflected a patient’s own tumor burden during neoadjuvant chemotherapy. Our earlier prospective study of >1600 individuals established a robust cut-off that cleanly separated healthy urine from urine from cancer patients with 90–95% specificity [27]; this normative benchmark is therefore already “built into” the assay. In the present trial, each participant serves as their own control; the IRS quantifies the directional change in the N-NOSE values between the baseline, preoperative, and postoperative samples. By focusing on these intra-individual trajectories, we remove the inter-subject variation in urine composition and isolate the biological effect of NAC on tumor-derived volatile cues. Consequently, longitudinal tracking of the N-NOSE index is sufficient to judge the therapeutic response, and the inclusion of an additional healthy control arm would offer no incremental analytical value while unnecessarily increasing the study complexity and cost.

Despite its novel findings, the current study is not free from limitations. First and foremost, it was conducted at a single institution with a small number of patients. Second, given that a CR was obtained only in HER2 or TNBC patients, the efficacy of N-NOSE screening in luminal-type breast cancer patients remains unclear. Thirdly, the reason for the differences in the AUCs between IRS1 and IRS3 remains a hypothesis and has not been conclusively proven. Lastly, it cannot be ruled out that differences in the urine collection conditions or the physiological state of *C. elegans* may have influenced the results.

## 5. Conclusions

Index reduction scores derived using the N-NOSE method may reflect the efficacy of NAC in breast cancer patients. Future large-scale and multi-institutional prospective studies are warranted.

## Figures and Tables

**Figure 1 cells-14-00950-f001:**
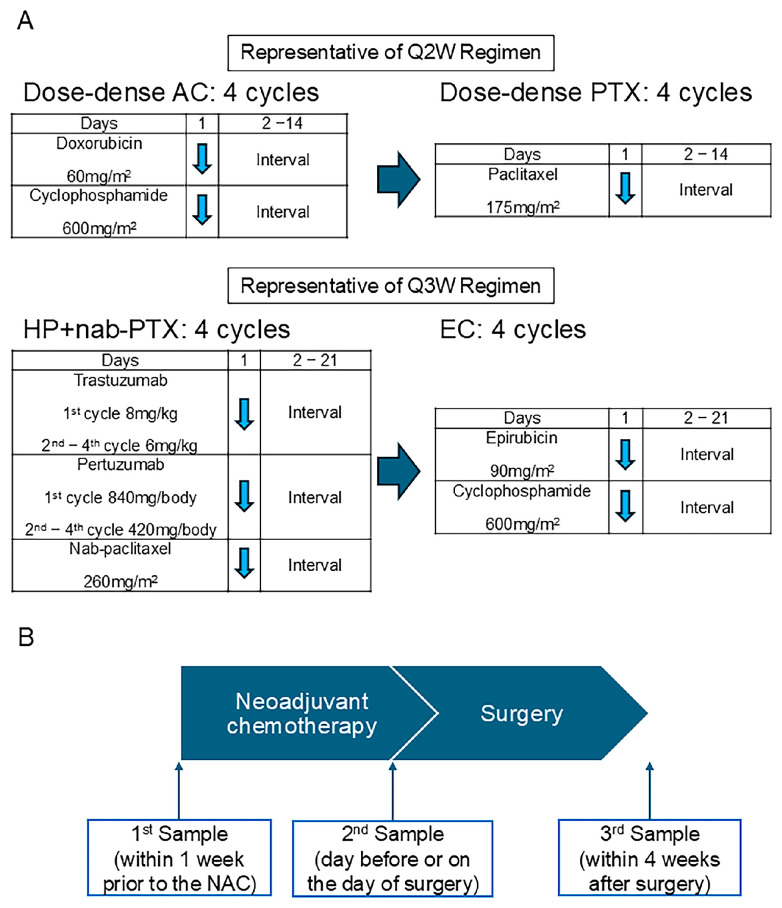
The neoadjuvant chemotherapy regimen and urine sample collection timeline. (**A**) Representative chemotherapy regimens q2W and q3W administered in the current study. (**B**) The scheme of the timing for urine sample collection. A: doxorubicin; C: cyclophosphamide; PTX: paclitaxel; nab-PTX: nab-paclitaxel; E: epirubicin; q2w: every 2 weeks; q3w: every 3 weeks. The blue arrow indicates the date on which chemotherapy was administered.

**Figure 2 cells-14-00950-f002:**
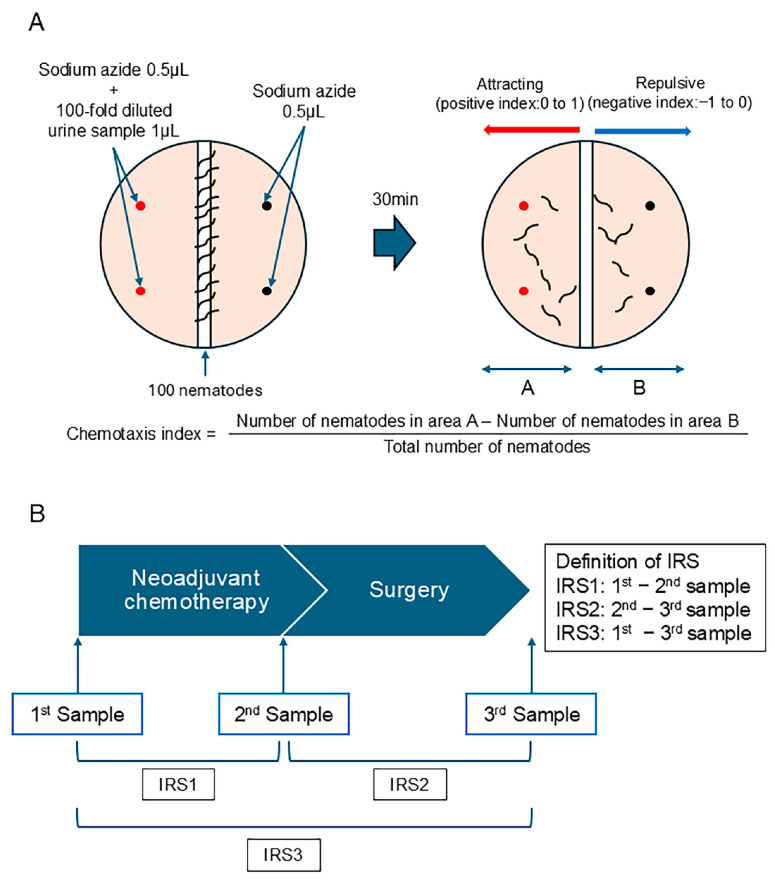
A scheme of the measurement method for the chemotaxis index and the definition of the index reduction score (IRS). (**A**) The framework for assessing the chemotaxis index. (**B**) The definition of the IRS.

**Figure 3 cells-14-00950-f003:**
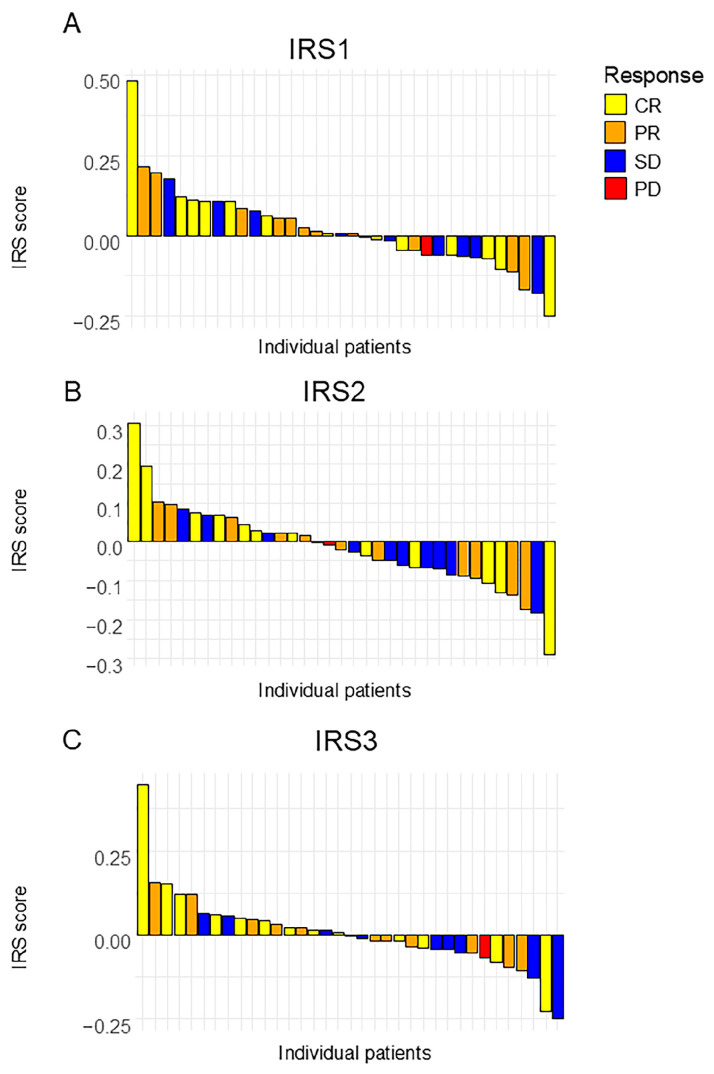
A waterfall plot of the analyzed index reduction scores. (**A**) IRS1, (**B**) IRS2, and (**C**) IRS3. Each bar represents an individual patient, sorted in descending order of score. CR: Complete Response; PR: Partial Response; SD: Stable Disease; PD: Progressive Disease.

**Figure 4 cells-14-00950-f004:**
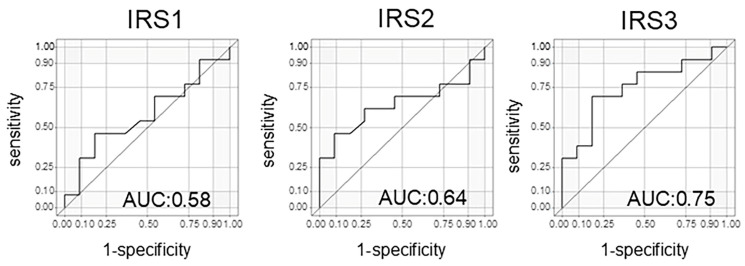
The area under the curve of the index reduction score (IRS) for patients with a CR only.

**Table 1 cells-14-00950-t001:** Clinical and pathological characteristics of the patients.

Clinicopathological Characteristics	Number of Patients (*n* = 36)
Age (median, range)	51 (35–77)
Menopause status	
Pre	17 (47.2%)
Post	18 (50.0%)
Unknown	1 (2.8%)
Clinical T stage	
1	4 (11.1%)
2	20 (55.6%)
3	5 (13.9%)
4	7 (19.4%)
Clinical N stage	
0	9 (25.0%)
1	17 (47.2%)
2	5 (13.9%)
3	5 (13.9%)
Clinical M stage	
0	36 (100.0%)
1	0 (0.0%)
Clinical stage	
I	1 (2.8%)
IIA	10 (27.8%)
IIB	11 (30.6%)
IIIA	5 (13.9%)
IIIB	4 (11.1%)
IIIC	5 (13.9%)
Subtype	
Luminal	11 (30.6%)
Luminal HER2	4 (11.1%)
HER2	10 (27.8%)
TN	11 (30.6%)
Treatment response	
CR	13 (36.1%)
PR	16 (44.4%)
SD	6 (16.7%)
PD	1 (2.8%)

**Table 2 cells-14-00950-t002:** Predictive accuracy of IRS 1, IRS2, and IRS for CR and PR patients.

	CR or PR				
	AUC	95%CI	Sensitivity	Specificity	Cut-Off Point
IRS1	0.53	0.19–0.87	0.79	0.40	0.11
IRS2	0.76	0.56–0.96	0.58	1.00	0.00
IRS3	0.66	0.37–0.96	0.79	0.6	−0.05

**Table 3 cells-14-00950-t003:** Predictive accuracy of IRS 1, IRS2, and IRS for CR patients only.

	CR				
	AUC	95%CI	Sensitivity	Specificity	Cut-Off Point
IRS1	0.58	0.34–0.82	0.46	0.82	0.06
IRS2	0.64	0.40–0.88	0.46	0.91	0.03
IRS3	0.75	0.54–0.95	0.69	0.82	0.00

## Data Availability

The data availability is restricted due to privacy and ethical restrictions.

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
