# Peer review of "Application of N-NOSE for Evaluating the Response to Neoadjuvant Chemotherapy in Breast Cancer Patients"

_cells, 2025, doi:10.3390/cells14130950_

Round 1

Reviewer 1 Report

Comments and Suggestions for Authors

This work is interesting.  The manuscript is well written. The reviewer has the following minor concerns.

  1. In Figure 1A: q2W and q3W regimen.

Please define q2W and q3W in the figure legend or in the main text.

  1. Page 5, line 154 -155: Statistical Analyses

In this sentence “Assuming a complete response (CR) rate of 30–50% and an expected ROC-AUC of 0.8 for the IRS”, please define ROC-AUC right after the term. (ROC-AUC: Area Under the Receiver Operating Characteristic Curve)

  1. For Table 3, the author should briefly describe how the values of Sensitivity and Specificity were determined from ROC curves in the Statistical Analyses of Materials and Methods.

Reviewer 2 Report

Comments and Suggestions for Authors

In this manuscript, Tokumaru et al., evaluate the application of N-NOSE in breast cancer patients. Previously, this group used the same tool for esophageal cancer patients (Sato et al., 2023, cancers) and extended the possibility of the application for other cancer types. Although this study has certain limitations, as the authors acknowledge in the Discussion, it offers potential insights into the extension of the N-NOSE method for non-invasive detection of diverse cancer types.

Minor comments

1. The authors described that sodium azide (used as an anesthetic) was added only to the side opposite the urine sample. This experimental setup could introduce bias, as C. elegans may freely roam on the urine side while being immobilized on the control side. Previous studies (Hirotsu et al., 2015, PLoS ONE; Sato et al., 2023, Cancers) applied sodium azide to both sides. The authors should clarify this point and justify or revise their methodology accordingly.

2. The Result section is too brief and lacks sufficient explanation of their results. The authors report numerical values without interpreting their significance. It would be helpful for readers if they explained their data in detail.

3. In Figure 3, the X-axis label is missing.

4. How "sensitivity" and "1-specificity" in Figure 4 and Table 2-3 were calculated, and what they mean, were not explained in the Materials and Methods or Results sections. The authors should provide clear explanations of how these values were derived.

5. The first three paragraphs in the Discussion section largely repeat content from the Introduction section. Consider revising this section.
